# *Mentha* Rhizomes as an Alternative Source of Natural Antioxidants

**DOI:** 10.3390/molecules25010200

**Published:** 2020-01-03

**Authors:** Silvia Bittner Fialová, Elena Kurin, Eva Trajčíková, Lucia Jánošová, Ivana Šušaníková, Daniela Tekeľová, Milan Nagy, Pavel Mučaji

**Affiliations:** 1Department of Pharmacognosy and Botany, Faculty of Pharmacy, Comenius University in Bratislava, Odbojárov 10, 832 32 Bratislava, Slovakia; elena.kurin@uniba.sk (E.K.); trajcikova1@uniba.sk (E.T.); susanikova1@uniba.sk (I.Š.); tekelova@fpharm.uniba.sk (D.T.); nagy@fpharm.uniba.sk (M.N.); mucaji@fpharm.uniba.sk (P.M.); 2Department of Pharmaceutical Analysis and Nuclear Pharmacy, Faculty of Pharmacy, Comenius University in Bratislava, Odbojárov 10, 832 32 Bratislava, Slovakia; veizerova@fpharm.uniba.sk

**Keywords:** *Mentha* rhizomes, antioxidant activity, interaction study, rosmarinic acid, lithospermic acid

## Abstract

Unlike its aerial parts, the underground parts of *Mentha* have so far been studied only marginally. By examining the polyphenolic fingerprint, the antioxidant efficacy and the mutual antioxidant behaviour of mixtures of mint rhizomes, our study presents a modest contribution to addressing this gap. Firstly, we examined the composition of the mint rhizomes: *Mentha × piperita* cv. ‘Perpeta’ (MPP), *M. longifolia* (ML), and *M. × villosa* cv. ‘Snežna’ (MVS). Our LC-MS-DAD analysis revealed the presence of ten compounds belonging to groups of phenolic acids and flavonoids, of which the rosmarinic acid (RA) and lithospermic were most strongly represented. Secondly, we evaluated the antioxidant activity of rhizome infusions by DPPH and ABTS and on NIH/3T3 cell lines by DCFH-DA. Thirdly, we determined, examined, and explained the mutual interactions of rhizome infusions mixtures. While most of the combinations acted additive, synergy was observed in ternary infusion mixtures. The synergic action was also detected in the combination of MPP rhizome infusion and RA in the DCFH-DA test. The combinations of mint rhizomes and rosmarinic acid displayed a high dose-reduction index. This leads to beneficial dose reduction at a given antioxidant effect level in mixtures, compared to the dose of the parts used alone. So far, the pharmaceutical and food industry has not used mint rhizomes in commercial products. Hence, our study draws attention to further applications of the *Mentha* rhizomes as a valuable alternative source of natural antioxidants.

## 1. Introduction

*Mentha* L. is a genus belonging to Lamiaceae family and Nepetoidae subfamily, which comprises around 18 species and an additional 11 hybrids placed into four sections: *Pulegium, Tubulosae, Eriodontes*, and *Mentha*. Mints are found worldwide [1]. Their aerial parts, especially leaves, contain important secondary metabolites, including essential oil (rich in monoterpenes and sesquiterpenes), rosmarinic acid, and flavonoids (such as eriodyctiol, luteolin, apigenin, and their glycosides) [2]. Due to their chemical content, mints are commercially established not only in the food industry, but owing to their olfactory properties, also in perfumery. Their further use include phytotherapy and various traditional medicinal systems [1,3].

Natural compounds, especially phenolic ones that possess antioxidant effects, are spread in many aromatic, spice, and medicinal plants. They serve as antioxidants in food, cosmetics, and other fields, where oxidation is undesirable [4]. Not unimportantly, several studies have confirmed that consumption of foods rich in antioxidants benefits human health [5].

Rosmarinic acid is a phenolic acid, which was first isolated by Scarpati and Oriente in 1958 [6]. It is richly present in species of the Lamiaceae family, along with aerial parts of mints. It possesses strong antioxidant properties due to being composed of two phenolic rings, which both contain two hydroxyl groups in *ortho*-position [7,8]. After oral administration, rosmarinic acid undertakes intensive pre-systemic metabolism and, consequently, only a small amount is detected in plasma [9]. Thus, research on rosmarinic acid interactions in natural extracts or artificial combinations may enrich its medicinal use by additional information on how to amplify its own bioavailability or final biological action.

Most species of the genus *Mentha* produce rhizomes. With the exception of peppermint propagation, they have not been known to be used as food, nor for medicinal or other purposes. A simple vegetative propagation of mints brings large plant material, including long, thin rhizomes [1]. When looking for studies dealing with secondary metabolites in mint underground parts, we found out that only a few focus on compounds in mints rhizomes (stolons). Mitchel et al. (1998) examined total non-structural carbohydrates by different methods to measure the energy stored in peppermint rhizomes in order to examine the regrowth ability of peppermint in terms of agriculture [10]. Karasawa and Shimizu (1980) detected triterpene acids in callus tissues in *Mentha arvensis* var. *piperascens* MAL [11]. The presence of monoterpenes and monoterpene glycosides in stolons was also described. Rhizomes can store and metabolize (catabolize) some volatiles created in aerial parts. Through the process of transportation from the peppermint leaves to rhizomes, the l-menthone is reduced to d-neomenthol and thereafter is glycosylated. Subsequently, it is reversely hydrolysed and oxidized to l-menthone and converted to l-3,4-menthone lactone [12,13]. What about polyphenols?

Medicinal plants and herbal medicinal products are poly-component mixtures. Due to this heterogeneity of constituents, they are present in numerous interactions [14]. Interactions are generally described as being synergistic or antagonistic. Being under synergy is “working together”, while under antagonism is “working against each other”. Between the two stays intermediate, zero-interactive state [15], commonly known as additivity. Searching for the synergistic composition of phytopharmaceuticals and discovering plants possessing such constituents, which lead to multitarget mode of actions, was known a long time ago in traditional medicine systems like Indian Ayurveda [16] or in TCM [17]. Nowadays, the capability of synergistic interaction is described also among antioxidants from natural sources, when mixed in specific combinations together [18,19] providing more efficient results. Mint rhizomes as a source of diverse antioxidants offer interesting extracts and compounds, whose interactions could be explored [20].

The aim of this study was to identify and quantify the main compounds in infusions of *Mentha* × *piperita* cv. ‘Perpeta’ (MPP), *Mentha* × *villosa* cv. ‘Snežná’ (MVS), and *Mentha longifolia* (ML) rhizomes. We evaluated the antioxidant activity of mint infusions in vitro and on cell culture and performed an interaction study of mint samples mixtures and their combinations with rosmarinic acid. To our best knowledge, this is the first time different *Mentha* rhizomes were examined in terms of polyphenol secondary metabolites and antioxidant properties.

## 2. Results and Discussion

In spite of various utility of *Mentha* aerial parts, mint rhizomes remain largely unexplored. Moreover, they lack proper application in the food and pharmaceutical industry. Fialová et al. researched the percentage content of total flavonoids and total hydroxycinnamic derivatives in methanol extracts of rhizomes of seven mint taxa [20]. They also compared the antioxidant activity of methanol extracts of rhizomes. The present research deals with phytochemical analysis and determination of antioxidant properties of mint rhizomes infusions.

### 2.1. Phenolic Fingerprint: Identification of the Characteristic Constituents

Phenolic compounds were identified by liquid chromatography mass spectrometry (LC-MS/MS), using authentic standards and database search by comparing the mass spectra with a maximum allowed mass deviation of 10 ppm. We detected the presence of ten phenolic compounds, phenolic acids, and flavonoids and one unknown compound (Appendix A), summarized in Table 1, along with their retention times (t_R_), observed mass in negative ionisation mode, MS/MS fragment ions of each secondary metabolite. The compounds detected in analysed samples were characterized by means of MS data, together with the interpretation of the observed MS/MS spectra in comparison with those listed in the available literature. The resolution of peak 4 and peak 10 presented complications. Peak 4 showed [M − H]^−^ at *m*/*z* 553, with the major fragment ions of at *m*/*z* 373 and *m*/*z* 329, and was identified as 2-(3,4-dihydroxyphenyl)ethyl ester of Salvianolic acid D. The fragment at *m*/*z* 535 from the pseudomolecular ion at *m*/*z* 715 was detected for peak 10, which might be attributed to the loss of caffeic acid (−180 mu). Further loss of 44 mu (carboxyl group moiety) from the ion at *m*/*z* 535 would release the fragment at *m*/*z* 491 (salvianolic acid C). Despite these observations, the definite structure could not be assigned, hence we described it only as caffeic acid tetramer. These compounds have been identified in *Mentha* species in aerial parts in the past [2,5,21,22,23], but this is for the first time, when different *Mentha* rhizomes were analysed. From Table 1 and Table 2, it is evident that the main constituents of *Mentha* rhizomes infusions are rosmarinic (54–142 μg/mL) and lithospermic acid (20–82 μg/mL).

### 2.2. Qualitative Determination of Constituents

The quantification of identified compounds was performed by using external standards. The proposed method was validated with the sensitivity and precision parameters. All standards showed good linearity. The following *r*^2^ values were obtained: protocatechuic aldehyde *r*^2^ = 0.9997, regression curve *y* = 594.9*x* − 81.401, LOD = 0.37 μg/mL, and LOQ = 1.24 μg/mL); caffeic acid *r*^2^ = 0.9998, regression curve *y* = 454.32*x* + 62.456, LOD = 0.46 μg/mL, and LOQ = 1.51 μg/mL); rosmarinic acid *r*^2^ = 0.9992, regression curve *y* = 182.1*x* − 19.432, LOD = 1.16 μg/mL, and LOQ = 3.89 μg/mL); luteolin-7-*O*-glucoside *r*^2^ = 0.9969, regression curve *y* = 99.007*x* − 267.46, LOD = 2.33 μg/mL, and LOQ = 7.77 μg/mL.

### 2.3. In Vitro Antioxidant Activity by DPPH and ABTS

Acting as a radical, 2,2-diphenyl-1-picrylhydrazyl (DPPH), and acting as a cation radical, 2,2′-azino-bis(3-ethylbenzothiazoline-6-sulfonic acid) diammonium salt (ABTS), are both well-known in in vitro scavenging assays based on electron transfer in the protic solvent [19]. Antioxidant activities of different *Mentha* sp. aerial parts and essential oils were previously published [24,25,26]. In this study, we measured the scavenging activity of MPP, ML, and MVS rhizomes in DPPH and ABTS assay. As can be seen in Figure 1, all infusions exhibited dose depending antioxidant activity, whereby ML had the highest activity in the DPPH model (IC_50_ 25.39 µg/mL), as well as in ABTS model (IC_50_ 21.30 µg/mL). MPP had similar antioxidant effects in DPPH (IC_50_ 28.45 µg/mL) and ABTS (IC_50_ 27.45 µg/mL) assay. MVS had the lowest activity in DPPH (IC_50_ 39.48 µg/mL) as well as in ABTS (IC_50_ 40.87 µg/mL). The antioxidant activity of rosmarinic acid is known [20,27]; therefore, it was used as a positive control and showed strong scavenging activity in DPPH (IC_50_ 2.28 µg/mL) and in ABTS (IC_50_ 1.05 µg/mL). When we calculated total mass from quantitative analysis of polar phenolic compounds in *Mentha* rhizomes infusions, we found out that MVS (122.8 ± 1.25 µg/mL) has significantly lower mass content (*p* < 0.05, ANOVA/Bonferroni) compared to MPP (185.5 ± 9.91 µg/mL) or ML (171.8 ± 1.17 µg/mL). Lower polar phenolic compounds content in MVS could enlighten its lower antioxidant activity in vitro. Fletcher et al. observed similar relation, that a positive correlation existed between total phenolic content and antioxidant capacity in different *Mentha spicata* plant clones leaves in the DPPH model.

### 2.4. Interaction Analysis of DPPH and ABTS

After single-infusion antioxidant-activity measurements, we tried to evaluate antioxidant activity of *Mentha* rhizomes in their equimolar binary and ternary mixtures, and their mixtures with rosmarinic acid. As can be seen in Table 3, mint-infusion combinations in DPPH and ABTS assays led mainly to additive interactions, except for ternary mixture of MPP:ML:MVS in ABTS assay, where slight synergism was noticed. Mint combinations with rosmarinic acid exhibit antagonistic interactions in both antioxidant assays. The study of Hajimehdipoor et al. showed that rosmarinic acid acted synergistically with caffeic acid in FRAP (ferric-reducing antioxidant power) model [28]. In another study, synergistic protective effects between rosmarinic acid and caffeic acid on 2,2′-azobis (2-amidinopropane) dihydrochloride (AAPH)-induced linoleic acid oxidation were observed. Authors showed that these mixture effects are partly explained by regeneration mechanisms between antioxidants, depending on the chemical structure of molecules and on the possible formation of stable intermolecular complexes [29]. However, the composition of our mint infusions is composed of numerous of compounds and could not be explained by simple two agents’ interaction. Although we observed antagonistic antioxidant effects between mints and rosmarinic acid, DRI (dose-reduction index) value was higher than 1 for each mint sample in mixtures with RA. By definition, the DRI value for a specific compound indicates, to what extent its concentration can be reduced in a mixture in order to achieve a given effect level, compared to the single treatment [30]. Interestingly, MVS infusion with the lowest single antioxidant activity had, in combination with rosmarinic acid, the highest DRI value. This means that the concentration of MVS can be reduced 14.80 times in DPPH, and 21.60 times in ABTS assay, when combined with RA compared to its single action. In the study of Fletcher et al., a direct correlation was found between rosmarinic acid content and the antioxidant capacity (*r* = 0.80) in different *Mentha spicata* clones in DPPH model [31]. We did not observed positive correlation between rosmarinic acid content and antioxidant activity in DPPH, but we need to consider bigger divergence in total content among three different *Mentha* species in comparison to single plant clones content. Likewise, artificial addition of rosmarinic acid to extracts decreases the IC_50_ value in equimolar mixtures rapidly, as it is seen in Table 3.

### 2.5. Intracellular Oxidative Stress Inhibition

DCFH-DA is a fluorogenic dye that measures hydroxyl radical, peroxyl one and other ROS activity within the cell. After diffusion into the cell, lipophilic DCFH-DA is deacetylated by cellular esterases to a non-fluorescent compound 2′,7′-dihydrodichlorofluorescein (DCFH), which is later oxidized by ROS into 2′,7′-dichlorofluorescein (DCF). DCF is a highly fluorescent compound, which can be detected by fluorescence spectroscopy with maximum excitation and emission spectra of 480 and 530 nm, respectively.

H_2_O_2_ is used as activator of intracellular oxidative stress. In our experiments, 15 min of incubation with H_2_O_2_ (100 µM) caused a 2-fold increase in ROS level toward the control cells (data not shown), which is comparable to previous studies [32,33].

The 1 h incubation of cells with different *Mentha* rhizomes infusions significantly decreased the intracellular ROS production in cells treated with H_2_O_2_, in comparision to the control (100%). Measurements were done in concentration range between 40 and 2.5 µg/mL of mint rhizomes samples, and their effects were compared to rosmarinic acid, which was measured in a concentration range between 40 and 0.15625 µg/mL. We observed concentration dependence in all examples. As it is seen in Figure 2, rosmarinic acid is the most potent examined compound, with IC_50_ 0.37 (*r* = 0.97) µg/mL. IC_50_ of MPP, ML, and MVS was measured as 9.21 (*r* = 0.99), 13.34 (*r* = 0.99), and 18.34 (*r* = 0.98) µg/mL, respectively. As the mass concentration of RA is approximately two times higher in MPP (115.9 ± 0.29 µg/mL) than in the MVS (55.7 ± 0.11 µg/mL), we can suppose that RA could be the major contributor to the total intracellular antioxidant activity in mint rhizomes samples. The mass concentration of RA is similar in ML (54.0 ± 0.14 µg/mL) and in MVS (55.7 ± 0.11 µg/mL), but IC_50_ of MVS is approximately 1.37-fold higher than IC_50_ of ML. The mass concentration of lithospermic acid is four times higher in ML (82.1 ± 0.12 µg/mL) than in MVS (20.6 ± 0.10 µg/mL). It indicates that lithospermic acid also plays a role in the total intracellular antioxidant activity in mint rhizomes samples.

DCF fluorescence indicates the resultant oxidative stress due to the overproduction of ROS or the depletion of antioxidants. It does not identify any specific ROS [34]. Lithospermic acid and RA have a number of phenolic hydroxyl groups, which are generally attributed to their powerful antioxidant properties. Antioxidant effect of lithospermic acid in vitro was previously described [35], as well as in in vivo models [36,37]. The antioxidant activity of rosmarinic acid in different cell lines was measured by DCFH-DA assay, several times before. According to Fernando et al., 2.5 µM RA scavenged 60% of intracellular ROS generated by H_2_O_2_ in HaCaT keratinocytes [38]. Ghaffari with co-workers observed a decrease in H_2_O_2_-induced ROS production in the N2A cells pretreated with RA in the concentration range 1–25 µM. Results were concentration-dependent [39]. The result of the study of Ramanauskiene et al. (2016) showed that the amount of ROS in C6 cells depends on the concentration of RA. Concentrations smaller than 150 µM RA reduced the number of intracellular radicals, and higher concentrations acted as prooxidants [40]. The results of all these studies confirm that, depending on the concentration, RA has an antioxidant effect and reduces the number of intracellular radicals in different cell lines, which is comparable to our results.

As we previously described, rosmarinic and lithospermic acids are polyphenols that are most abundantly present in tested *Mentha* rhizomes. We confirmed in vitro antioxidant effect of RA. Antioxidant effect of lithospermic acid in vitro was previously described, as well as in in vivo models [35,36,37]. Because of that, both acids contribute to the antioxidant efficacy of used extracts.

### 2.6. Interaction Analysis of Intracellular Oxidative Stress Inhibition

We performed an interaction analysis of *Mentha* rhizomes equimolar samples mixtures with rosmarinic acid in intracellular oxidative stress inhibition assay, using the same mathematical method as described above with in vitro assays. As it can be seen in Figure 3, IC_50_ of mint rhizomes is rapidly decreased when combined with RA. Interaction analysis confirmed synergy in MPP:RA combination, whereas ML:RA and MVS:RA mixtures were antagonistic, similar to results in DPPH and ABTS assays. DRI value [30] in Table 4 shows that the concentration of MPP can be reduced 28.74 times, ML 14.77, and MVS 25.79 times when combined with RA compared to its single action.

We confirmed the synergistic effect of MPP:RA mixture at 50% effect dose level with the combination index calculation, and this result can be also seen in the illustrative isobologram (Figure 3). From the isobologram definition, the solid line segment is based on the IC_50_ of both compounds individually and indicates the collection of concentration pairs that have no interaction, i.e., hypothetical additive action. Points under the solid line are the dose pairs which elicit the same effect with a smaller dose in total and therefore are synergistic, whereas points above the solid line are the dose pairs which elicit the same effect with a higher dose and thus are antagonistic [14]. The point in the middle of the curved dashed line in Figure 3 shows the experimentally found position of the IC_50_ value of the MPP and RA mixture and manifests the synergistic interaction of this mixture. As we mentioned above, MPP rhizome sample has the highest content of RA from the examined mints and shows the highest intracellular oxidative stress inhibition. The addition of RA can synergistically an increase its own activity probably due to increase of total RA concentration in the mixture.

## 3. Materials and Methods

### 3.1. Plant Material

The original specimens of *M.* × *piperita* cv. ‘Perpeta’ (MPP) and *M.* × *villosa* cv. ‘Snežná’ (MVS) were bought from Seva Flora (Valtice, Czech Republic). *M. longifolia* (ML) was collected in wild nature (Modra-Zochová chata, district Pezinok, Slovakia, 48°22′48.4′′ N 17°16′33.2′′ E). The identity of the investigated plants was verified according to the literature [41], using anatomical, morphological, and phytochemical parameters. All rhizomes (around 500 g of each) were collected in the autumn time (October), after the second flowering. The plant material was dried at room temperature (25 °C), in the drying room. Voucher specimens were deposited at the Department of Pharmacognosy and Botany, Faculty of Pharmacy, Comenius University, in Bratislava, Slovakia.

### 3.2. The Preparation of Infusions

The infusions of rhizomes were prepared according to the Czech–Slovak Pharmacopoeia, 4th edition [42]. For the preparation of infusion, powdered rhizome (sieve 355) was moistened with the fivefold amount of water and soaked for 15 min. Afterward, the remaining boiling water was added to the rhizome, kept 5 min on a water bath, and rested at room temperature for another 45 min. Prepared infusions were filtered and lyophilized.

### 3.3. LC-MS-DAD Analyses: Identification and Quantification of the Characteristic Constituents

The LC-MS analyses were performed on an Agilent 1260 Infinity LC System (Agilent Technologies, Santa Clara, CA, USA), equipped with a binary pump, an autosampler, a column thermostat, and a diode array detector (DAD), coupled to a quadrupole-time of flight (6520 Accurate-Mass QTOF) instrument equipped with an orthogonal electrospray ionisation source (ESI) (Agilent Technologies, Santa Clara, CA, USA). HPLC separation of mint infusions was carried out on a Kromasil C18 column (4.6 × 150 mm, 5 µm, Sigma-Aldrich, Munich, Germany), at 35 °C, and a flow rate of 0.4 mL/min. Water (pH 3.1 with HCOOH/NH_4_HCO_2_) and acetonitrile were used as mobile phase A and B, respectively. The following gradient program was used: 10% B (20 min), 20% B (25 min), 60% (50 min), 95% (62 min), and 10% (70 min). The ESI ion source parameters were as follows: capillary voltage = 3.5 kV; nebulizer = 40 psi (N_2_); dry gas flow = 10 L/min (N_2_); and dry temperature = 300 °C. The mass spectrometer was operated in an autoMS^2^ mode, where each negative ion MS scan (*m*/*z* 100–3000, average of 4 spectra) was followed by MS^2^ scans (*m*/*z* 100–3000, average of 4 spectra, isolation window of 4 amu, collision energy 20 eV) of the two most intense precursor ions. Ions were excluded from analyses for 0.5 min, after two MS^2^ spectra were acquired. Nitrogen was used as the collision gas. Phenolic compounds were identified by comparing their UV and mass spectra with the literature and authentic standards, if available, and by measuring accurate *m*/*z* [27]. Used authentic standards (protocatechuic aldehyde, caffeic acid, eriodictyol-7-*O*-rutinoside, rosmarinic acid, hesperetin-7-*O*-rutinoside, salvianolic acid B, luteolin-7-*O*-rutinoside) were purchased by Sigma-Aldrich (St. Louis, MO, USA).

The quantitative determination of phenolic compounds in *Mentha* rhizomes infusions was provided by the method of external standards. We used analytical standards of protocatechuic aldehyde (purity 97%, Sigma-Aldrich, St. Louis, MO, USA) for determination of protocatechuic aldehyde, caffeic acid (purity ≥ 98.0%, Sigma-Aldrich, St. Louis, USA) for the quantification of caffeic acid, luteolin-7-*O*-glucoside (purity ≥ 98.0%, Sigma-Aldrich, St. Louis, MO, USA) for the quantification of flavonoid glycosides, and rosmarinic acid (purity 96%, Sigma-Aldrich, St. Louis, MO, USA) for the rest of phenolic acids and their derivatives (see Table 2). The examinations of secondary metabolites in mints rhizomes were performed in triplicate. The quantitative results were calculated from calibration curves and expressed as mean values and standard deviation (*SD*).

### 3.4. Antioxidant Activity by DPPH and ABTS

DPPH Scavenging Assay: DPPH radical scavenging assay was determined according to Blois (1958), with slight adaptations [43]. Briefly, stable 2,2-diphenyl-1-picrylhydrazyl (DPPH, Sigma-Aldrich, St. Louis, MO, USA) radical was ex tempore dissolved in methanol. DPPH (225 µL, 55 µM) was mixed with mint sample dissolved in deionized water or rosmarinic acid dissolved in ethanol (25 µL, 15.63 µg/mL–1 mg/mL). Changes in absorbance were measured in 96-well Greiner UV-Star microplates (Greiner-Bio One GmbH, Frickenhausen, Germany), with a Tecan Infinite M200 microplate reader (Tecan AG, Grödig, Austria), after 30 min, at 517 nm. All measurements were done in quadruplicate.

ABTS Scavenging Assay: ABTS radical scavenging assay was determined according to Re et al. (1999), with slight adaptations [44]. Briefly, 7 mM of 2,2′-azinobis-(3-ethylbenzothiazoline-6-sulfonic acid) (ABTS, purity ≥ 98%, Sigma-Aldrich, St. Louis, MO, USA) aqueous solution was mixed in equimolar ratio with 2.45 mM of K_2_S_2_O_8_ (purity ≥ 99.0, Sigma-Aldrich, St. Louis, MO, USA), allowing the mixture to stand in the dark, at room temperature, for 24 h before use. After radical stabilization, the ABTS solution was diluted with ethanol (1.1 mL), to a final volume of 50 mL. Mint sample dissolved in deionized water or rosmarinic acid dissolved in ethanol (2.5 µL, 1.56 µg/mL–0.1 mg/mL) was added to the ABTS solution (247.5 µL). Changes in absorbance were measured in 96-well Greiner UV-Star microplates (Greiner-Bio One GmbH, Germany), a with Tecan Infinite M200 microplate reader (Tecan AG, Austria), after 6 min, at 734 nm. All measurements were done in quadruplicate.

### 3.5. Intracellular Antioxidant Activity Testing

Biological material/cell culture: NIH/3T3 (mouse embryonic fibroblasts) were obtained from the Department of Pharmacology and Toxicology, Faculty of Pharmacy, Comenius University, in Bratislava, Slovakia. The cells were grown at 37 °C, in humidified air, with 5% CO_2_ in DMEM (Sigma-Aldrich, St. Louis, MO, USA) supplemented with 10% FBS (Sigma-Aldrich, St. Louis, MO, USA), 100 IU/mL penicillin (Sigma-Aldrich, St. Louis, MO, USA), and 100 µg/mL of streptomycin (Sigma-Aldrich, St. Louis, MO, USA). The cells were passaged approximately twice a week. Detection of oxidative stress with DCFH-DA (purity ≥ 97%, Sigma-Aldrich, St. Louis, MO, USA) was determined according to Miranda-Rottmann et al. (2003), with slight adaptations [33]. The cells NIH/3T3 were seeded in the serum-free medium into a Nunc PP black 96-well plate (15,000 cells/100 µL/well) and were allowed to grow for 24 h. After 1 h of incubation, the medium was changed by the fresh serum-free medium. After 1 h, tested samples (5 µL) were added. Then, 5 µL of DCFH-DA dissolved in the serum-free medium (a final concentration 5 µg/mL) was added after 1 h of incubation. After 15 min, 5 µL of H_2_O_2_ were added, with a final concentration 100 µM. The intracellular fluorescence of dye DCF was measured after 15 min, with an excitation/emission wavelength 480/530 nm, and compared to a blank by using a Tecan Infinite M200 (Tecan AG, Austria) microplate reader. Results were expressed as IC_50_ (µg/mL) and calculated by CompuSyn 1.0.1 (ComboSyn, Inc., Paramus, NJ, USA) software. All measurements were done in quadruplicate.

### 3.6. Interaction Analysis

The interaction analysis evaluating synergy or antagonism of the combinations was done according to the mass-action law principle [30], described by Equation (1) for *n*-drug combination at *x*% inhibition, using a combination index (CI) for interaction interpretation.
(1)$${}^{n}{\left(\mathrm{CI}\right)}_{x}=\;\overset{n}{\underset{j=1}{\sum }}\;{\left(D\right)}_{j}/{\left(D_{x}\right)}_{j}$$

^n^(CI)_x_ is the sum of the dose of n drugs that exerts x% inhibition in a combination. In the denominator (D_x_) is for D “alone” that inhibits a system x%. If CI value is =, > or < 1, an additive, synergistic or antagonistic effect is indicated. Using the log(CI) grading, synergism and antagonism can be subdivided by Chou [30] into 11 ranges of CI, as seen in Table 5.

The dose-reduction index (DRI) expresses how many times the dose of each drug in a combination could be reduced at a given effect level, compared with the doses of each drug alone. The DRI value for each corresponding drug was given for n-drug combinations, as illustrated in Equation (2):(2)$${\left(\mathrm{DRI}\right)}_{1}=\frac{{\left(D_{x}\right)}_{1}}{{\left(D\right)}_{1}};\;{\left(\mathrm{DRI}\right)}_{2}=\frac{{\left(D_{x}\right)}_{2}}{{\left(D\right)}_{2}}\ldots \mathrm{etc}.$$

The value of DRI > 1 indicates a favourable dose reduction, and the higher DRI value indicates the higher dose reduction for a given therapeutic effect, but does not necessarily always indicate synergism. Both CI and DRI, as well as concentration leading to 50% inhibition (IC_50_), were calculated by using a median-effect analysis by CompuSyn software.

## 4. Conclusions

Although the medicinal and food usage of peppermint underground parts (rhizomes) is currently unknown, this research shows their pharmaceutical and food industry potential as an alternative source of natural antioxidants, especially of rosmarinic and lithospermic acid. The results of the antioxidant effects of infusions of three mint rhizomes show efficacious activity and synergic behaviour of selected combinations. All this knowledge may lead to the usage of mint rhizome in applications such as stabilizers or preservatives in the food and as a ROS modulator in medicine.

## Figures and Tables

**Figure 1 molecules-25-00200-f001:**
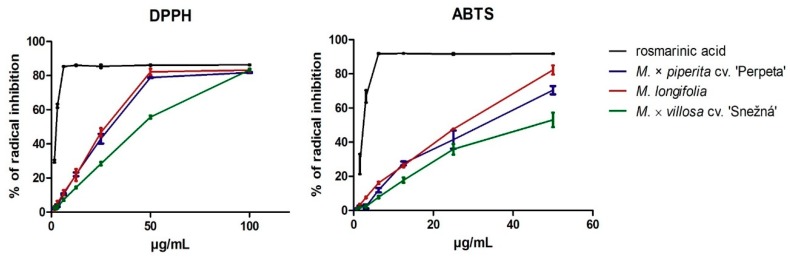
Dose-dependent DPPH and ABTS radical inhibition of single mint extracts and rosmarinic acid. The bars represent mean ± *SD*, *n* = 4.

**Figure 2 molecules-25-00200-f002:**
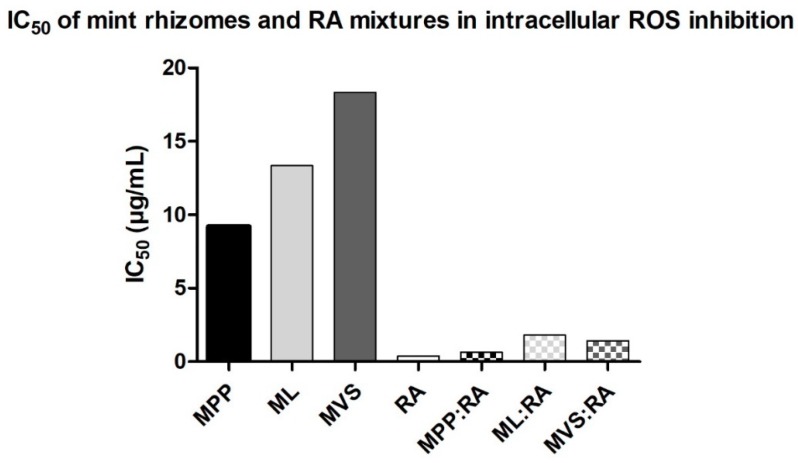
IC_50_ (µg/mL) of *Mentha* rhizomes extracts (MPP—*Mentha* × *piperita* cv. ‘Perpeta’, ML—*Mentha longifolia*, MVS—*Mentha* × *villosa* cv. ‘Snežná’, RA—rosmarinic acid, and their mixtures in the intracellular ROS inhibition in NIH/3T3 cells, using DCFH-DA assay. IC_50_ and *r* values (MPP = 0.99, ML = 0.99, MVS = 0.98, MPP:RA = 0.97, ML:RA = 0.95, MVS:RA = 0.96) were calculated by using CompuSyn software, *n* = 4. *r* value: the conformity parameter for the goodness of fit to the median-effect principle (MEP) of the mass-action law. It is the linear correlation coefficient of the median effect plot, where *r* 1 indicates perfect conformity.

**Figure 3 molecules-25-00200-f003:**
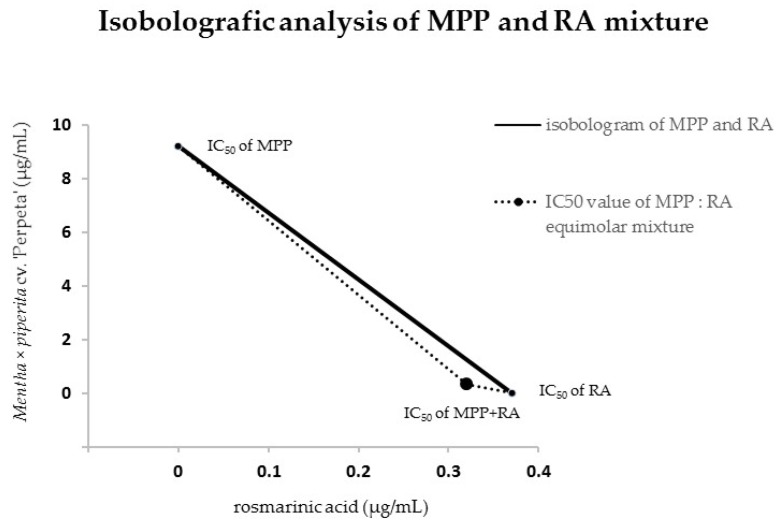
Illustrative isobologram of MPP:RA mixture at 50% effect dose level, (MPP—*Mentha* × *piperita* cv. ‘Perpeta’; RA—rosmarinic acid).

**Table 1 molecules-25-00200-t001:** Polar phenolic compounds in infusions of *Mentha* rhizomes, their corresponding retention times (t_R_), molecular ions [M − H], and MS^2^ fragments in LC-MS analysis.

Peak	Compound	t_R_ (min)	[M − H]^−^ (*m*/*z*)	MS^2^ (20 eV) (*m*/*z*)
1.	Protocatechuic aldehyd	17.59	137	
2.	Caffeic acid	26.67	179	
3.	Eriodictyol-7-*O*-rutinoside	35.29	595	287
4.	2-(3,4-dihydroxyphenyl)ethyl ester of Salvianolic acid D	41.14	553	373, 329, 197, 179
5.	Rosmarinic acid	45.28	359	197, 161
6.	Hesperetin-7-*O*-rutinoside	47.84	609	301
7.	Salvianolic acid B	48.16	717	519, 321, 179
8.	Lithospermic acid	48.64	537	359
9.	Salvianolic acid A	49.83	493	359, 313
10.	Caffeic acid tetramer	50.30	715	535, 491
11.	Luteolin-7-*O*-rutinoside	50.89	593	285

**Table 2 molecules-25-00200-t002:** Quantitative abundance of polar phenolic compounds in infusions of *Mentha* rhizomes (μg/mL).

Compound	Mass Concentration (μg/mL) * ± *SD*		
	*M. × piperita* cv. ‘Perpeta’	*M. longifolia*	*M. × villosa* cv. ‘Snežná’
Protocatechuic aldehyde ^a^	<LOQ	nd	<LOQ
Caffeic acid ^b^	2.8 ± 0.02	nd	<LOQ
Eriodictyol-7-*O*-rutinoside ^c^	nd	14.4 ± 0.82	nd
2-(3,4-dihydroxyphenyl)ethyl ester of Salvianolic acid D ^d^	7.1 ± 3.18	nd	4.8 ± 0.04
Rosmarinic acid ^d^	115.9 ± 0.29	54.0 ± 0.14	55.7 ± 0.11
Hesperetin-7-*O*-rutinoside^c^	19.3 ± 0.29	nd	8.9 ± 0.45
Salvianolic acid B ^d^	14.6 ± 0.07	21.3 ± 0.09	9.0 ± 0.01
Lithospermic acid ^d^	25.8 ± 0.06	82.1 ± 0.12	20.6 ± 0.10
Salvianolic acid A ^d^	nd	nd	4.0 ± 0.16
Caffeic acid tetramer ^d^	nd	nd	10.7 ± 0.10
Luteolin-7-*O*-rutinoside ^c^	<LOD	nd	9.1 ± 0.28
Total flavonoids	20.8 ± 0.69	14.4 ± 0.82	18.0 ± 0.17
Total phenolic acids	166.7 ± 1.12	157.4 ± 0.35	106.3 ± 0.59
Total phenolics	187.5 ± 0.43	171.8 ± 0.47	124.3 ± 0.41

* Values (μg/mL liquid infusion) are presented as means ± standard deviation (*n* = 3), external standards: protocatechuic aldehyde ^a^; caffeic acid ^b^; luteolin-7-*O*-glucoside ^c^; rosmarinic acid ^d^; LOQ-limit of quantification; LOD—limit of detection.

**Table 3 molecules-25-00200-t003:** Interaction analysis of mutual mint infusions and mint infusions/rosmarinic acid combinations in DPPH and ABTS assays: IC_50_, CI, and DRI values of equal mass concentration infusions combinations at 50% inhibition dose level.

	Infusions Combination ^a^	IC_50_ (μg/mL) ^b^	*r*	CI ^c^	SDA ^d^	Combined Effect	DRI ^e^
DPPH	MPP:ML	**25.95** (28.45:25.39)	0.98 (0.99:0.98)	0.97	±0.02	nearly additive	2.19:1.96
	MPP:MVS	**33.38** (28.45:39.48)	0.99 (0.99:0.99)	1.01	±0.01	nearly additive	1.70:2.37
	ML:MVS	**33.82** (25.39:39.48)	0.97 (0.98:0.99)	1.09	±0.02	nearly additive	1.50:2.33
	MPP:ML:MVS	**31.68** (28.45:25.39:39.48)	0.99 (0.99:0.98:0.99)	1.05	±0.01	nearly additive	2.69:2.41:3.74
	MPP:RA	**7.49** (28.45:2.28)	0.98 (0.99:0.99)	1.78	±0.03	antagonism	7.60:0.61
	ML:RA	**11.01** (25.39:2.28)	0.98 (0.98:0.99)	2.63	±0.05	antagonism	4.61:0.41
	MVS:RA	**5.33** (39.48:2.28)	0.98 (0.99:0.99)	1.24	±0.02	moderate antagonism	14.80:0.85
ABTS	MPP:ML	**21.83** (27.45:21.30)	0.98 (0.97:0.98)	0.91	±0.03	nearly additive	2.52:1.95
	MPP:MVS	**44.07** (27.45:40.81)	0.98 (0.97:0.99)	1.34	±0.04	moderate antagonism	1.25:1.85
	ML:MVS	**29.21** (21.30:40.81)	0.98 (0.97:0.99)	1.04	±0.03	nearly additive	1.46:2.79
	MPP:ML:MVS	**24.84** (27.45:21.30:40.81)	0.96 (0.97:0.98:0.99)	0.89	±0.03	slight synergism	3.32:2.57:4.93
	MPP:RA	**3.64** (27.45:1.05)	0.98 (0.97:0.99)	1.80	±0.03	antagonism	15.09:0.58
	ML:RA	**4.48** (21.30: 1.05)	0.99 (0.98:0.99)	2.24	±0.03	antagonism	9.51:0.47
	MVS:RA	**3.78** (40.81:1.05)	0.98 (0.99:0.99)	1.85	±0.03	antagonism	21.60:0.56

^a^ MPP—*Mentha* × *piperita* cv. ‘Perpeta’; ML—*Mentha longifolia*; MVS—*Mentha* × *villosa* cv. ‘Snežná’; RA—rosmarinic acid. ^b^ Median inhibitory activities IC_50_ (μg/mL) of the equal mass concentration infusions combinations and in bracket their single infusion/compound IC_50_ level. ^c^ CI—combination index, a quantitative measure based on the mass-action law of the degree of drug interaction in terms of synergism (CI < 1) and antagonism (CI > 1) for a given endpoint of the effect measurement. The combined effect is evaluated according to Chou (2006). ^d^ SDA-sequential deletion analysis, iterative sequential deletion of one dose (or concentration) of a drug at a time for repetitive CI calculations. ^e^ DRI represents the order of magnitude (fold) of dose reduction that is allowed in combination for a given degree of effect as compared with the dose of each drug alone.

**Table 4 molecules-25-00200-t004:** Interaction analysis of mutual mint infusions and mint extracts/rosmarinic acid combinations in DCFH-DA assays: IC_50_, CI, and DRI values of equal mass concentration infusions combinations at 50% inhibition dose level.

	Infusions Combination ^a^	IC_50_ (μg/mL) ^b^	*r*	CI ^c^	SDA ^d^	Combined Effect	DRI ^e^
DCFH-DA	MPP:RA	0.64 (9.21:0.37)	0.97 (0.99:0.97)	0.89	±0.04	slight synergism	28.74:1.17
	ML:RA	1.81 (13.34: 0.37)	0.95 (0.99:0.97)	2.49	±0.10	antagonism	14.77:0.41
	MVS:RA	1.42 (18.34:0.37)	0.96 (0.98:0.97)	1.94	±0.08	antagonism	25.79:0.53

^a^ MPP—*Mentha* × *piperita* cv. ‘Perpeta’; ML—*Mentha longifolia*; MVS—*Mentha* × *villosa* cv. ‘Snežná’; RA—rosmarinic acid. ^b^ Median inhibitory activities IC_50_ (μg/mL) of the equal mass concentration infusions combinations and in bracket their single infusion/compound IC_50_ level. ^c^ CI—combination index, a quantitative measure based on the mass-action law of the degree of drug interaction in terms of synergism (CI < 1) and antagonism (CI > 1) for a given endpoint of the effect measurement. The combined effect is evaluated according to Chou (2006). ^d^ SDA-sequential deletion analysis, iterative sequential deletion of one dose (or concentration) of a drug at a time for repetitive CI calculations. ^e^ DRI represents the order of magnitude (fold) of dose reduction that is allowed in combination for a given degree of effect as compared with the dose of each drug alone.

**Table 5 molecules-25-00200-t005:** The scales of combined effects by using the log(CI) grading.

	Range of Combination Index	Description
**Synergism**	<0.1	Very strong synergism
	0.1–0.3	Strong synergism
	0.3–0.7	Synergism
	0.7–0.85	Moderate synergism
	0.85–0.90	Slight synergism
	0.90–1.10	Nearly additive
**Antagonism**	1.10–1.20	Slight antagonism
	1.20–1.45	Moderate antagonism
	1.45–3.3	Antagonism
	3.3–10	Strong antagonism
	˃10	Very strong antagonism

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
