# Peer review of "Mentha Rhizomes as an Alternative Source of Natural Antioxidants"

_molecules, 2020, doi:10.3390/molecules25010200_

Round 1

Reviewer 1 Report

This article provides an interesting source of information about constituents and antioxidant activity of Mentha rhizomes.

Introduction part is appropriate and contains adequate information about this study.

The results are exactly described, in terms of their importance, statistical and interaction analyses are presented. Graphic presentation of the results and description of figures is precise. Sometimes “in vitro” is not in cursive, especially in part “2.3 In vitro antioxidant activity by DPPH and ABTS”. Discussion of the results is satisfactory. I have comments about identification of compounds 3, 6 and 11. Identification was performed by comparison with standards or only by analysis of MS spectra?  How was identified position of sugar at flavonoid glycosides included in Table 1?

I have comments about title of part 2.5 Intracellular oxidative stress. Maybe it would be better to use title of part 2.5 Intracellular oxidative stress inhibition

Methods used for the extraction, LC-MS analysis and antioxidant analysis of extracts and components are correct.

Conclusions are clear

References are sufficient and coherent to the study. No comments are necessary.

This manuscript is suitable for publishing in its current form with only minor revision.

Author Response

Dear reviewer, thank you for the time you paid to review our manuscript, as well as for your very positive feedback to our research, we appreciated a lot. Here below please find the information about the changes we made according to your valuable suggestions:

 Sometimes “in vitro” is not in cursive, especially in part “2.3 In vitro antioxidant activity by DPPH and ABTS”.

Corrected

I have comments about identification of compounds 3, 6 and 11. Identification was performed by comparison with standards or only by analysis of MS spectra?  How was identified position of sugar at flavonoid glycosides included in Table 1?

Yes, authentic standards were used for all three flavonoids (Rha-Glu in position 7). This additional information was implemented in the main text (Materials and Methods). 

I have comments about title of part 2.5 Intracellular oxidative stress. Maybe it would be better to use the title of part 2.5 Intracellular oxidative stress inhibition

Done

Reviewer 2 Report

The presented paper is interesting,  but it has some very significant shortcomings. While I have no objections to the first part (phenolic compound profile and antioxidant activity), studies on the interaction of active compounds raise great doubts.It is not clear how the authors determined CI. This value is mathematically determined from the data obtained during the isobolographic analysis. Testing of only one mixture is not enough to obtain reliable data. For the isobolographic analysis, the activity of several mixtures in different ratios should be tested. From the course of the curve presented in Figure 3 it is not clear whether the components of the mixture really interact synergistically. There is no statistical analysis in the whole work.

Author Response

Dear reviewer, thank you for the time you paid to review our manuscript, as well as for all of your suggestions on how to improve its text. The information about the changes we made and the explanations of our intention follows:

Interaction analysis in this article was performed by using CompuSyn 1.0.1 software (as we mentioned in the 4.6 Interaction analysis section), which used a combination index (CI) as a mathematical basis for its calculations. We described CI mathematically in Eq. 1. CI term was introduced by Chou and Talalay in 1983 for the quantification of synergism or antagonism for two drugs (Chou and Talalay, 1983, 1984; Chou, 1991). The review article by Chou (2006) describing CI has till today 2448 Scopus citations, which gives us reason to use this method as a plausible interaction analysis. As we wanted to be conformal with this method, we have adopted a whole protocol of CI calculation, as it is described by Chou. The base is to gain experimentally data for the dose-response curve for single agents and their mixture in a fixed ratio. IC50 (median effect) is then calculated from this dependence using CompuSyn software. Here we want to remark, that as Chou writes, for even a single data point (from a single combination dose) of a drug combination mixture, the CI value can still be calculated by Eq. 1 (Chou, 2006) as we did in our work. The combination index is then calculated as the sum of quotients of doses of agents in mixture and doses of single agents at a given effect level (50% in our case). If CI value is =, > or < 1, an additive, the synergistic or antagonistic effect is indicated. Statistical analysis was performed by the CompuSyn software as well. For the IC50 was done using r value: the conformity parameter for the goodness of fit to the median-effect principle (MEP) of the mass-action law. It is the linear correlation coefficient of the median effect plot, where r 1 indicates perfect conformity (Chou 2006). For CI value, sequential deletion analysis (SDA) was performed. SDA: an iterative sequential deletion of one dose (or concentration) of a drug at a time for repetitive CI calculations. This is followed by calculating the mean ± 95% confidence interval at each specified effect level of the Fa-CI plot (Chou 2006). R-values and SDA for calculated IC50 and CI are present in tables 3 and 4. Isobologram in our article is illustrative (we have added this note to the article already), as Chou (2006) described, that isobologram (ED50 isobol, ED75 isobol, ED90 isobol, etc.) is graph indicating the equipotent combinations of various doses of two drugs. It can be used to illustrate the additive effect, synergism, or antagonism, at different dose levels. CI value (used in our isobologram) with SDA for the RA:MPP mixture at 50% effect level is given as 0,89 ± 0,04 (as could be seen in table 4). This fulfills the synergy definition, where CI should be lower than 1.

Chou TC and Talalay P (1983) Analysis of combined drug effects: a new look at a very old problem. Trends Pharmacol Sci 4:450–454.

Chou TC and Talalay P (1984) Quantitative analysis of dose-effect relationships: the combined effects of multiple drugs or enzyme inhibitors. Adv Enzyme Regul 22: 27–55.

Chou TC (1991) The median-effect principle and the combination index for quantitation of synergism and antagonism, in Synergism and Antagonism in Chemotherapy (Chou TC and Rideout DC eds) pp 61–102, Academic Press, San Diego.

Chou TC (2006) Theoretical Basis, Experimental Design, and Computerized Simulation of Synergism and Antagonism in Drug Combination Studies. Pharmacol Rev 58:621–681, 2006.

Reviewer 3 Report

In the present manuscript, the polyphenolic fingerprint, the antioxidant efficacy and the mutual antioxidant behaviour of mixtures of mint rhizomes were evaluated.

Specific comments

Lines 52-54: rephrase sentence.

Line 62: do not use Italics for the term "var.".

Line 77: correct to "provide more efficient results".

Table 2: add extra lines at the end of the Table for total phenolic acids, total flavonoids and total phenolic compounds content.

Quality of figures is low.

Line 163: DRI should be defined when first cited in the text.

The scales used for combined effect characterization should be defined in the M&M section in an extra Table, as presented by Chou 2006.

Materials and Methods

How many plants were collected for each species. What was the number of biological samples (repetitions) for each analysis? 

Delete Lines 369-380.

Author Response

Dear reviewer, thank you very much for the time you paid to review our manuscript, as well as for all of your suggestions on how to improve its text. We made the following changes:

Lines 52-54: rephrase sentence.

Rephrased

Line 62: do not use Italics for the term "var.".

Done

Line 77: correct to "provide more efficient results".

Done

Table 2: add extra lines at the end of the Table for total phenolic acids, total flavonoids and total phenolic compounds content.

Done

Quality of figures is low.

The quality of Fig 1 was improved.

Line 163: DRI should be defined when first cited in the text.

Done

The scales used for combined effect characterization should be defined in the M&M section in an extra Table, as presented by Chou 2006.

Done

8. How many plants were collected for each species.

The information was added to the Material and Methods… around 500 g of each mint. It is difficult to specify the number of plants, because one rhizome may produce more stems/aerial parts.

9. What was the number of biological samples (repetitions) for each analysis? 

It is already mentioned in each analysis in section Material and Methods.

10. Delete Lines 369-380.

Done

Reviewer 4 Report

Estimate total phenolic and flavonoid content in the extract. Perform FRAP assay for detecting the antioxidant capacity of the extract. Provide chromatography result image in the result section. 2 to 40 µg/mL concentration of extract (MPP, ML and MVS) were used for determining the ROS inhibition in NIH/3T3 cell line. On what basis the dosage was chosen? 1-hour extract treatment to the cells was enough for evaluating the ROS inhibition ability of the extract? What was the effect of cell viability of NIH/3T3 cells on (0.2 to 40 µg/mL concentration) MPP, ML and MVS treatment? IC50 should not inhibit the cell viability. Discussion section need to be improved. Line 369-380. Irrelevant to the manuscript. Overall English correction is needed.

Author Response

Dear reviewer, thank you for the time you paid to review our manuscript, as well as for all of your suggestions on how to improve its text. The information about the changes we made and the explanations of our intention follows:

1. Estimate total phenolic and flavonoid content in the extract.

Done

2. Perform FRAP assay for detecting the antioxidant capacity of the extract.

We decided not to perform FRAP assay, because when an interaction analysis is evaluated by the combination index (Chou 2006), normalization of measured data is necessary for IC50 value calculation from the dose-response curve. This would lead in the case of FRAP assay to an additional artificial intervention. Concurrently, FRAP and similar Fe(III) reduction-based ET assays produce Fe(II) as the reduction product, which could give rise to the generation of reactive species (such as hydroxyl radicals) upon Fenton-like reactions with H2O2, thereby causing “redox cycling” of phenolics and yielding erroneous results. Moreover, when metal ion-containing probes (such as those of FRAP) are used, strong chelating or complexing agents (like o-dihydroxy derivatives present in our extracts) may interfere with the assay. These all were arguments not to perform a FRAP assay.

Chou TC (2006) Theoretical Basis, Experimental Design, and Computerized Simulation of Synergism and Antagonism in Drug Combination Studies. Pharmacol Rev 58:621–681, 2006.

3. Provide chromatography result image in the result section.

All chromatograms were added as supplementary material Figure S1 – mentioned in the main text.

4. 2 to 40 µg/mL concentration of extract (MPP, ML and MVS) were used for determining the ROS inhibition in NIH/3T3 cell line. On what basis the dosage was chosen?

For determining the ROS inhibition in NIH/3t3 cell line we used 2.5 to 40 µg/mL concentration of rhizomes extract, and 0.15625 to 40 µg/mL concentration of rosmarinic acid. We corrected it in the manuscript. The dosage was chosen according to results obtained by DPPH and ABTS methods. From previous experiments we knew, the IC50 in ABTS, DPPH and DCFH-DA would be in similar concentration range. For easier weighting and diluting we chose the above-mentioned concentration range.

5. 1-hour extract treatment to the cells was enough for evaluating the ROS inhibition ability of the extract?

For the short-term studies, it is enough. We studied the ability of extracts to inhibit ROS formation/effect at the moment. If we wanted to study another mechanism of antioxidant activity (for example induction/inhibition of specific enzymes), we would definitely choose longer treatment.

6. What was the effect of cell viability of NIH/3T3 cells on (0.2 to 40 µg/mL concentration) MPP, ML and MVS treatment? IC50 should not inhibit the cell viability. Discussion section need to be improved.

We provided no viability tests in the mention concentration range because after 1-hour preincubation by these low concentrations we do not expect any viability changes.

7. Line 369-380. Irrelevant to the manuscript.

Done

8. Overall English correction is needed.

Done

Round 2

Reviewer 2 Report

The presented paper is suitable for publication in its present form.

Author Response

Dear reviewer, thank you again for the time you paid to review our manuscript and its revised version.

Reviewer 4 Report

English language need to be improved.

Author Response

Dear reviewer, thank you again for the time you paid to review our manuscript and its revision. We again did the overall English correction.